# Contimask: Explaining Irregular Time Series Models via Perturbations in Continuous Time

**Max Moebus, Björn Braun, and Christian Holz**
Department of Computer Science, ETH Zurich
Zurich, Switzerland
{max.moebus};{bjoern.braun};{christian.holz}@inf.ethz.ch

## Abstract

Explaining black-box models for time series data is critical for the wide-scale adoption of deep learning techniques across domains such as healthcare. Recently, explainability methods for deep time series models have seen significant progress by adopting saliency methods that perturb masked segments of time series to uncover their importance towards the prediction of black-box models. Thus far, such methods have been largely restricted to regular time series. Irregular time series, however, sampled at irregular time intervals and potentially with missing values, are the dominant form of time series in various critical domains (e.g., hospital records). In this paper, we conduct the first evaluation of saliency methods for the interpretation of irregular time series models. We first translate techniques for regular time series into the continuous time realm of irregular time series and show under which circumstances such techniques are still applicable. However, existing perturbation techniques neglect the timing and structure of observed data, e.g., informative missingness when data is not missing at random. Thus, we propose **Contimask**, a simple framework to also apply non-differentiable perturbations, such as simulating that parts of the data had not been observed using NeuroEvolution. Doing so, we successfully detect how structural differences in the data can bias irregular time series models on a real-world sepsis prediction task where 90% of the data is missing. Source code is available on GitHub.

## 1 Introduction

Deep learning promises to change the analysis of time series data across various domains, including healthcare, economics, ecology, or physics. To enable the broad-scale application of deep learning models in these domains, model internals have to be verifiable. Suggestions made by models must become explainable for humans using them, which has become the challenge of the field of explainable AI ($X$AI). Especially in fields such as healthcare, where models could be used to assist patient treatment, human-level explainability is vital to enable impactful broad-scale application [9].

Explainability methods for deep learning initially surged for image models [6, 38, 31], and have seen some recent progress for time series [13, 14, 4, 5, 22]. Methods for explaining time series models initially translated masking and perturbation techniques from the image domain to the realm of time series [4, 5] and extended them to create more realistic perturbations that more naturally imitate time series [14]. So far, these methods have only focused on regularly sampled time series, i.e., samples arrive at equally spaced intervals and without missing values.

However, in many domains, time series data arrive at irregularly sampled intervals since data is often only generated in response to a specific event. For example, in healthcare, the physician and the state of the patient's health determine when data is collected so that hospital resources are not wasted [9]. Similarly, in mobile and wearable computing, signals might only be partially

39th Conference on Neural Information Processing Systems (NeurIPS 2025).

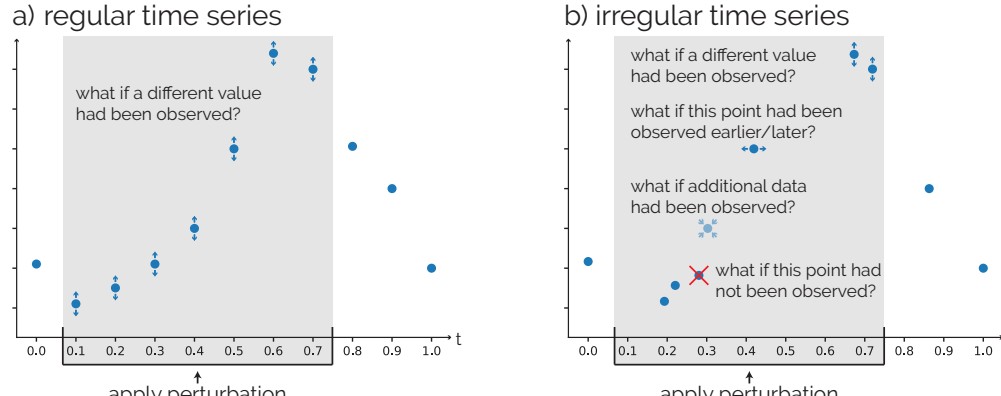

Figure 1: **a)** for regular time series, the timing of observations is fixed. Counterfactual examples thus simulate different values for existing timestamps. **b)** for irregular time series, counterfactual examples can take many different forms. They might simulate different values for existing timestamps, change the timing of observations, remove observations all together or simulate additional observations. **Contimask** adopts methods from regular time series to simulate different values of observed data. We further introduce a perturbation to simulate that data had not been observed, which helps to uncover the importance of time intensity and missing data, which is neglected by current methodologies.

measurable [16, 18, 17, 1]. In such scenarios, data is not missing at random, and missing data and the time interval between data collection can be informative. Irregular time series models are capable of not only considering observed data, but also informative missingness (also called time intensity) [8]. These models showed particular success for modeling hospital and general practitioner (GP) records, where up to 90% of data might be missing, and data is generally sparse [8, 9, 33].

In scenarios such as the ones described above, it is of particular importance to verify model internals. Take the example of hospital records: does the model simply predict that a patient will have a certain disease because a physician ordered a related test? In this case, the model would be of little use in practice. However, if information collected way before the treating physician's diagnosis is already contributing towards the model's prediction, the model might be a useful aid to improve patient care.

Logically, since the timing of observations is fixed for regularly sampled time series, existing masking and perturbation techniques to explain time series models ignore the timing of observations (see Figure 1). This poses an issue when trying to explain models for irregular time series when data is not missing at random, such as in the cases described above.

In this paper, we investigate techniques to calculate saliency maps for irregular time series models. We identify scenarios where adopting perturbation techniques for regular time series work well for irregular time series, and conditions under which they fail to recover the true saliency. We then propose **Contimask**: a framework applying non-differentiable perturbations to irregular time series data to uncover also saliency connected to the timing/missingness of data. **Contimask** operates in the continuous time realm of irregular time series and utilizes NeuroEvolution to train masks that, besides altering observed data points, simulate that data had not been observed.

**Contributions**    To the best of our knowledge, Contimask is the first method to calculate saliency maps for irregular time series models without access to model internals and to apply non-differentiable perturbations that alter the timing and structure of data. Doing so, we are able to explain also models where missingness influences predictions—common for irregular time series. In summary, we

- propose a novel perturbation for irregular time series that simulates that data has not been observed,

- propose a novel method that uses NeuroEvolution to apply also non-differentiable perturbations,

- show that small networks in continuous time are more effective to calculate saliency maps for irregular time series—especially if one incorporates Fourier feature activations, and

- showcase on a real-world prediction task how irregular time series models pick up structural biases.

## 2 Background & Related Works

### 2.1 Irregular Time Series

Irregular time series differ from regular time series in that observations are not collected at uniform time intervals, and not all channels of the time series have to be observed at all times, i.e., missingness. This format is prevalent in many real-world applications such as healthcare, where data is collected in response to events rather than on a fixed schedule, and not always the same information is collected.

Understanding and modeling these gaps is critical for downstream tasks. Consequently, any framework for interpreting or predicting from irregular time series must account not only for the observed values but also for the timing and structure of data (and the structure of missing data).

**Notation**   To formalize the setting of irregular time series, we unify well-established notation [8] that ultimately represents the $n^{\text{th}}$ element of a dataset $\mathcal{D}$ as a triplet $(t_n, x_n, d_n)$. We assume a supervised setting, where we have some black-box function $f$ that maps elements of $\mathcal{D}$ to labels $y$. Therefore, $\mathcal{D} = \{(s_n, y_n) \mid n = 1, \ldots, N\}$ is a dataset with $N$ elements. An individual element has a label $y_n$ and a sparse and irregularly sampled multivariate time series with $C$ channels, say $s_n$.

The $C$ channels of the multivariate time series $s_n$ might be sampled at different times, leading to varying numbers of total observations per channel: $L_{cn}$. Each channel $c$ of time series $s_n$ can thus be noted as $s_{cn} = (t_{cn}, x_{cn})$, where $t_{cn} = [t_{cn}^1, \ldots, t_{cn}^{L_{cn}}]$ and $x_{dn} = [x_{dn}^1, \ldots, x_{dn}^{L_{dn}}]$ are the list of time points and corresponding observations of channel $c$ of element $\{s_n, y_n\} \in \mathcal{D}$. Ultimately, to transform the time series into a triplet of tensors $(t, x, d)_n = (t_n, x_n, d_n)$, we let $t_n = \bigcup_{c=1}^{C} t_{cn} \in \mathbb{R}^{|t_n|}$ contain the sorted unique time stamps across all channels. We let $x_n \in \mathbb{R}^{|t_n| \times C}$ combine all observed values across all $C$ channels, and $d_m$ be a binary data mask indicating if channel $c$ has been observed:

$$x_n(i, c) = \begin{cases} x_{cn}^j & \text{if } t_{cn}^j = t_n^{(i)} \\ \texttt{nan} & \text{otherwise} \end{cases} \quad , \quad d_n(i, d) = \begin{cases} 1 & \text{if } \exists j \text{ such that } t_{cn}^j = t_n^{(i)} \\ 0 & \text{otherwise} \end{cases} .$$

**Model Architectures**   Almost all models for irregular time series involve some level of smoothing or interpolation—either on raw data or derived hidden representations. These models are designed to handle the non-uniform sampling and potential missingness inherent in irregular time series. Earlier techniques used Gaussian Process Adapters [12], or learned both high- and low-frequency smoothing functions [26]. Oftentimes, such approaches would later discretize the time axis again and simply extract representations interpolated in continuous time at some fixed reference points [12]. Currently, the majority of irregular time series models fall into two main architectural families:

**Continuous-time recurrent units:**   These include Neural Ordinary, Controlled, and Stochastic Differential Equations (`NODEs` [2], `NCDEs` [8], and `NSDEs` [20]). These approaches generalize discrete recurrent neural networks (RNNs) and model hidden states in continuous time using differential equations, or using a continuous-time Kalman filter [24].

**Encoder-decoder models with multi-time attention:**   Multi-time attention (`mtan` [28]) leverages attention mechanisms that operate across irregular timestamps. `mtan` [28], and adoptions thereof, have been particularly successful at interpolating irregular time series [3, 27, 37].

### 2.2 Post-hoc Instance-Level Explanation

In this paper, we attempt what has been termed "Post-hoc Instance-level Explanation" [14]. Originally concerned with image classification, such methods have the goal of explaining the prediction of an arbitrary black-box model $f$ for a single input: e.g., explaining why the provided image of a dog is indeed classified as a dog [29]. The output of such methods is often described as a saliency map [38]. Such maps highlight the parts of the input that are deemed important towards the prediction [31].

The earliest explainability methods tracked representations and/or gradients through the consecutive hidden layers of the network [38, 29, 31], many requiring access to intermediate layers or architectural modifications [6]. This was followed by approaches that modified parts of the input, observing changes in the prediction of $f$ to detect areas of importance [6]. Many of the current explainability methods for regular time series focus on the objective introduced by Fong and Vedaldi [6]:

For simplicity, we will ignore the previously defined irregular time series triplet for now, and note any model inputs as $x$, i.e., we aim to explain some black-box $f : \mathcal{X} \rightarrow \mathcal{Y}$. Further, let $m \in [0,1]^{\dim(x)}$ be a soft mask and $\Phi(x, m)$ a perturbation of $x$. Then, Fong and Vedaldi [6] optimize the following objective (excluding some further penalty terms on $m$):

$$\min_m \mathcal{L}(f(x), f(\Phi(x, m))) + \lambda \|\mathbf{1} - m\|_1, \tag{1}$$

where $\mathcal{L}$ is some loss function encouraging large perturbations that only minimally affect $f(x)$.

### 2.2.1 Explanations for Regular Time Series

Most works on time series explainability are based on the objective defined by Fong and Vedaldi [6], where $m$ is learned via gradient descent due to the problem being differentiable (given that $f$ is itself differentiable). Earlier works are `FIT` [36] and `DynaMask` [4], who also introduced crucial artificial benchmark problems. While `FIT` tracks the predictive distribution of $f$ over time to estimate feature importance, `DynaMask` translates the objective defined in Eq. 1 to the realm of time series. Motivated by Fong and Vedaldi [6], Crabbé and Van Der Schaar [4] introduce a penalty on the total variation of $m$, encouraging smoothness and optimizing for $m$ to cover a set proportion $a \in (0, 1)$ of $x$.

$$\min_{m \in [0,1]^{T \times c}} \mathcal{L}(y, f(\Phi(x, m))) + \lambda_1 \|\text{vecsort}(m) - r_a\|^2 + \lambda_2 \sum_{t=1}^{T-1} \sum_{i=1}^{C} |m_{t+1,i} - m_{t,i}|, \tag{2}$$

where $x \in R^{T \times C}$ is a fully observed time series of length $T$ with $C$ channels, $m \in [0,1]^{T \times C}$, $r_a$ is a sorted vector with the same number of entries as $m$ with the desired proportion $a$ being equal to 1 (otherwise 0), which encourages sharp masks that cover proportion $a$ of $x$.

`Dynamask` proposes two main perturbations for time series: a Gaussian blur $\Phi^{GB}$ and fading to a moving average $\Phi^{FMA}$. Both perturbations are applied independently for each channel $c$:

$$\Phi^{FMA}(x, m)_{t,c} = m_{t,c} \cdot x_{t,c} + (1 - m_{t,c}) \cdot \mu_{t,c}, \tag{3}$$

$$\Phi^{GB}(x, m)_{t,c} = \frac{\sum_{t'=1}^{T} x_{t',c} \cdot g_{\sigma(m_c)}(t - t')}{\sum_{t'=1}^{T} g_{\sigma(m_{t,c})}(t - t')}, \tag{4}$$

where $g_\sigma(t) = \exp\left(-\frac{t^2}{2\sigma^2}\right)$ with $\sigma(m) = \sigma_{\max} \cdot (1 - m)$, $\mu_{t,c} = \frac{1}{2W+1} \sum_{t'=t-W}^{t+W} x_{t',c}$, and $W \in \mathbb{N}$ is a sliding window size.

Works like `ExtremalMask` [5], `TimeX++` [13], and `ContraLSP` [14], have replaced closed-form perturbations by small RNNs to learn perturbations $\Phi(x, m, \theta)$, where $\theta$ parametrizes a RNN. This led to perturbations imitating the original data more closely and improved the resulting saliency maps.

## 2.3 Explainability Methods for Irregular Time Series

Currently, there exists no method for post-hoc instance-level irregular time series explanation. Even though `TimeX` [22], a self-supervised pre-training strategy to detect explanations via consistency constraints in the latent space, can be adopted to work for methods such as `mtan`, it requires access to model internals. Further, it does not work for architectures such as `NCDE` due to the CDE solve.

## 2.4 NeuroEvolution

Gradient descent is the predominant optimization technique to 'learn' the parameters of neural networks, say $f$. Techniques such as Adam [10] or Stochastic Gradient Descent (SGD) [23] require $f$ to be differentiable, and the objective $f$ is trying to optimize using $f$ to be differentiable also. NeuroEvolution on the other hand, does not require $f$ or the objective to be differentiable and thereby is capable of solving a much larger set of problems [7]. While NeuroEvolution is evidently slower, and at times, less reliable than gradient descent, recent work demonstrated state-of-the-art MNIST30K performance using NeuroEvolution [11]. In comparison to Neuroevolution, Adam and SGD operate at light speed. However, they require the objective to supply gradients, i.e., to be differentiable.

# 3 Methods

Our approach **Contimask** makes a step towards post-hoc instance-level explanations for irregular time series models to also detect saliency dependent on missingness and time intensity. Such methods alter, i.e., perturb, the input based on some learned mask $m$ and observe changes in model output. Loosely speaking, if part of the input can be altered greatly without changing the model output, such a region is considered uninformative (i.e., non-salient) towards the prediction of the model. Conversely, if small changes in some regions have a large effect on model output, such regions would be deemed informative (i.e., salient) towards the prediction of the model. All current approaches for explaining time series models only change the value of observed data and ignore the timing/presence of data as a possible explanation. For irregular time series, the action space of producing viable counterfactual examples is much larger than for regular time series (see Figure 1). Since for regular time series the timing of data is fixed (and thereby the fact that data was observed), only the value of observed data can be altered. A counterfactual example of irregular time series might include different timing of data points, different values of observed data points, data might be missing altogether, or additional data might have been observed. We strive that our approach generalizes to any black box function $f$ and does not require access to or specific knowledge of the model internals.

**Finding a saliency map in continuous time**  Following notation from [6, 4, 5], we optimize the following objective for some black-box model $f$ that takes as input a triplet $(t_n, x_n, d_n) \in \mathcal{D}$ as defined in Section 2.1. Assume we are trying to explain $f(t_n, x_n, d_n)$, where $|t_n| = T$, and $x_n \in \mathbb{R}^{T \times C}$ is a time series with $C$ channels, and therefore $d_n \in [0, 1]^{T \times C}$, where $x_n$ might include missing values that are marked with a 0 in $d_n$, non-missing entries are denoted by 1 in $d_n$. In contrast to prior work, we define $m : \mathbb{R} \to [0, 1]^C$ as a function operating in continuous time. Provided some perturbation $\Phi$, and a loss function $\mathcal{L}$, we optimize:

$$\min_m \mathcal{L}(f(t_n, x_n, d_n), f(\tilde{t}_n, \tilde{x}_n, \tilde{d}_n)) + \lambda_1 \sum_C \int_0^T (1 - m(u)_c) du + \lambda_2 \sum_C \int_0^T |m(u)'_c| du, \quad (5)$$

where $(\tilde{t}_n, \tilde{x}_n, \tilde{d}_n) = \Phi((t_n, x_n, d_n), m)$. Similarly to Eq. 1, we are trying to maximize the area covered by $m$ while ensuring $m$ is somewhat smooth and minimizing changes in model output.

We adopt $\Phi^{GB}$ and $\Phi^{FMA}$ as used in `DynaMask` to the setting of irregular time series, i.e., where $m$ is a function in continuous time and $x$ includes missing values as indicated by the binary mask $d$:

$$\Phi^{\text{FMA}}\big((t, x, d), m\big)_{i,c} = \left(t_i, \begin{cases} m(i)_c \, x_{i,c} + (1 - m(i)_c) \, \mu_{i,c} & \text{if } d_{i,c} = 1, \\ x_{i,c} & \text{if } d_{i,c} = 0, \end{cases} , d_{i,c}\right), \quad (6)$$

$$\Phi^{\text{GB}}\big((t, x, d), m\big)_{i,c} = \left(t_i, \begin{cases} \dfrac{\sum_{t'=1}^T x_{t',c} \cdot g_{\sigma(m(t)_c)}(i - t')}{\sum_{t'=1}^T g_{\sigma(m(t)_c)}(i - t')} & \text{if } d_{i,c} = 1, \\ x_{i,c} & \text{if } d_{i,c} = 0, \end{cases} , d_{i,c}\right). \quad (7)$$

where $g_\sigma(i) = \exp\left(-\frac{i^2}{2\sigma^2}\right)$ with $\sigma(m) = \sigma_{\max} \cdot (1 - m)$ and $\mu_{i,c} = \frac{1}{2W+1} \sum_{i'=i-W}^{i+W} x_{i',c}$, where $W \in \mathbb{N}$ is a sliding window size. Note that $t$ and $d$ remain unchanged, and only the value of the observed data $(x)$ is altered. Please find visual examples of these perturbations in Appendix A.1.

**A perturbation that alters the structure of observed data**  We further introduce a simple, yet effective, data perturbation that removes data points, i.e., simulates that data points had not been observed. We refer to this perturbation simply as the `Deletion`-perturbation—`Del` in short. This forms the first perturbation to also affect $d$ of our triplet (and potentially $t$):

$$\Phi^{\text{Del}}\big((t, x, d), m\big)_{i,c} = \begin{cases} (t_i, x_{i,c}, d_{i,c}) & \text{if } Ber(m(i)_c) = 0 \\ (t_i, \texttt{nan}, 0) & \text{if } Ber(m(i)_c) = 1 \end{cases} \quad (8)$$

Here, $Ber(p)$ denotes the sample from a Bernoulli distribution with probability of success equal to $p$. If $Ber(m(i)_c) = 1$, `Del` therefore alters $x$ and $d$. Effectively, in case there are time points for which all data is being removed, $t$ is also being altered. We ignore this scenario in the above notation.

**Non-differentiability of perturbations that alter the structure of data** Without knowledge of the specific model internals, simulating that data had not been observed is non-differentiable w.r.t. the applied perturbation mask $m$, since this requires $m$ to be a hard mask (i.e., a binary or boolean mask). This holds true for all perturbations that affect the structure of the data, i.e, that affect not just $x$, but also $d$, or the order of $t$, With access to and exact knowledge of model internals, deletion of data can be simulated for `mtan` models or, partially, also for models such as `NCDE` (see Appendix A.2). Perturbations that alter the timing of data or insert new data points (see Fig. 1) will face similar issues. Therefore, Contimask relies on gradient-free optimization to learn mask $m$.

We learn $m$ using NeuroEvolution and employ the PGPE algorithm [25, 7] as implemented in EvoTorch [35] using the ClipUp optimizer [34]. While NeuroEvolution gets around the issue of non-differentiability, it trains much slower, which we discuss more extensively in Appendix A.3.

**Perturbation masks as functions in continuous time** We define saliency, and the mask we are trying to construct in continuous time. Rather than optimizing for a tensor $m \in \mathbb{R}^{T \times C}$, we learn $m(t) : \mathbb{R} \to [0,1]^C$ as a small feedforward neural network. We show that this reduces the required number of parameters to accurately define $m$ and, particularly when employing NeuroEvolution, leads to better performance and faster training. To learn $m$, we evaluated various small network architectures. Beyond a 3-layer `MLP` with ReLu activations, we evaluated architectures that incorporated feature transformations to create sharper masks. In particular, we tested sinusoidal representations (SIRENs) [30], Haar functions [21], and Fourier feature transformations [32]. We ultimately opted for Fourier feature transformations given performance on initial baselines (Table 1) and proven success on images [15]. In this paper, `MT` implies $m \in \mathbb{R}^{T \times C}$, whereas `MLP` implies $m(t) : \mathbb{R} \to [0,1]^C$.

**MLPs with and without Fourier feature transformations** We term the small `MLP` with Fourier transformations `MLP-F`. Given a scalar input $t$, `MLP-F` applies a fixed encoding using $L$ exponentially spaced Fourier frequencies, transforming the input into a $2L$-dimensional feature vector composed of sine and cosine terms. If not stated otherwise, we set $L = 10$, such that:

$$\gamma(t) = \left[ \sin(2^k \pi t), \cos(2^k \pi t) \right]_{k=0}^{L-1}.$$

These features, $\gamma(t)$, are then passed through a `MLP` with three linear layers and ReLU activations, mapping to an output of dimension $C$. When we refer simply to `MLP`, we feed a scalar input $t$ straight into three linear layers with ReLU activations. We use a final sigmoid layer such that $m(t) \in [0,1]^C$.

## 4 Evaluation

We consider 5 problem settings. We first convert two commonly used synthetic scenarios for regular time series explanations into the continuous time setting. Termed 'Rare Time' and 'Rare Feature' by Crabbé and Van Der Schaar [4], they depend on a low number of salient times and a low number of salient features, respectively, and form scenarios that saliency methods have traditionally struggled with [22, 13, 14]. The original versions of these two artificial problems depend on the value of observed data. We will refer to this setting as **value-based**, i.e., the function that we are trying to explain depends on the element $x$ of our triplet defined in Section 2.1. We then adapt these two scenarios and create two problem settings, where the value of observations (i.e., $x$) does not impact the output of the function we are trying to explain, but only the timing of data, i.e., $t$. We refer to this setting as a **temp-based**. We finish by explaining a model trained on a common problem for irregular time series models: sepsis prediction from hospital records based on [8].

**Rare Time & Rare Feature** In these two artificial problems, we create white-box regressors that only depend on a salient area, say $A = A_T \times A_X$, where $A_X \subset [1 : C]$. While in the regular time series settings, $A_T \subset [0 : T]$, we define $A_T \subset [0, 1]$ again in continuous time.

In the Rare Time scenario, we consider 100 randomly sampled time points in the interval [0,1] and 3 channels. At each iteration of the experiment, we sample a random location for each of the three channels to define our salient area. The salient area covers 20% of the first channel, 10% of the second, and 45% of the third. In the Rare Feature scenario, we sample again 100 time points but simulate a time series with 50 channels. For each iteration, we randomly sample 5 consecutive features that are the only salient ones, and we set $A_T = [0.25, 0.75]$ in all cases.

For both scenarios, as per [4], the output of the white-box regressor is $\sum_{(t,c)\in A} x_{t,c}^2$.

**Temp-based settings**  We adapt the previously defined scenarios and change the white-box regressor to $\sum_{(t,c)\in A} 1$, i.e., we count all data points in $A$ such that the output is only dependent on $t$.

**Sepsis prediction**  We train a `NCDE` and `mtan` model on the sepsis prediction task as implemented by Kidger et al. [8]. We then take these models, and apply the `Del`, `FMA`, and `GB` perturbations to train `MLP-F` masks to cover 10% of the observed data to explain two cases: one where the model is initially confident that a patient will die of sepsis within the first 72 hours in the hospital; and one where the model is initially confident of survival. We only explain cases on the test set (5.4% mortality). In addition to static variables (i.e. fixed over time), 34 features were sampled irregularly over 72 hours at 1 hour resolution with $\approx 90\%$ missing values.

Since the `NCDE` model trained by Kidger et al. [8] takes as input cubic spline coefficients rather than the irregular time series triplet (Section 2.1), we first have to reconstruct the test set from the spline coefficients. At each iteration, we then apply our perturbations on the reconstructed data, calculate the new cubic spline coefficients, and feed those into the model to observe the change in prediction. Recalculating the coefficients for a single evaluation is very slow (30 s on H200 GPU). Therefore, we train smaller `MLP-F` masks with a hidden dimension of 16, and $L = 12$ Fourier features for only 200 epochs using PGPE and limit ourselves to two participants with low and high probability of sepsis. While the `mtan` model operates directly on the irregular time series and is therefore much faster to explain, we stick to above settings for fair comparability. vram

**Metrics**  For the Rare Time & Rare Feature settings, ground truth saliency maps are available. We calculate the F1 score (F1), Precision (Prec), and Recall (Rec) for correctly identifying these maps, using a threshold of 0.5 to binarize masks. However, masks are usually already binary, as also indicated by very low $S_m$ values, especially for `MLP-F` masks. Further, we calculate metrics introduced in [4]: $I_m(A) = -\sum_{(i,c)\in A} \ln(1 - m(i)_c)$ to capture the information content of mask $m$ over the salient area $A$ (higher is better), and $S_m(A) = -\sum_{(i,c)\in A} m(i)_c \ln m(i)_c + (1 - m(i)_c)\ln(1 - m(i)_c)$ to measure the sharpness of the mask over $A$ (lower is better).

For sepsis prediction, we calculate the average change in log-odds as the main metric. To allow for a fair comparison between `Del`, which removes data, and `FMA` and `GB`, which only alter the value of observed data, we calculate two metrics for each mask $m$. First, we apply `Del` based on the output of $m$ to observe the change in predictions, which we term '`Del` odds change'. Second, we impute 0 for all data points suggested by the trained mask, similar to [4, 5], which we term '`Imp` odds change'. Note that as per [8], all features are normalized, i.e., imputing 0 inserts the mean feature.

**Experiment setup**  We run all experiments using an H200 GPU needing at most 8GB of VRAM.

## 5   Results

In Table 1 we show how gradient descent and gradient-free training via NeuroEvolution compare for the value-based Rare Feature problem when applying the `FMA` perturbation. Here, we compare different mask parameterizations, all with similar parameter count close to 5,000. We set $\lambda_1 = 0.01, \lambda_2 = 0.001$ and train for 16,000 epochs using an Adam optimizer with a learning rate of 0.01, or 2000 iterations using the PGPE optimizer with a population size of 100. For PGPE, we initialize with a radius of 3, and a center learning rate of 0.5 ($\pm 0.3$). While `MLP` and `MLP-F` have not yet meaningfully converged after 16,000 iterations, `MT` achieves an almost perfect F1 score using gradient descent. However, when training using NeuroEvolution, `MLP` and `MLP-F` outperform `MT`, even outperforming the gradient descent training. We provide a more detailed comparison between the different training strategies, and how parameter count influences convergence and computation cost in Appendix A.3. Since `MLP-F` consistently outperforms `MLP`, we chose `MLP-F` in all subsequent problems with a hidden layer size 32 and $L = 10$, resulting in a parameter count of 3,400.

In Table 2, we compare different perturbations to train masks parameterized by `MLP-F` using NeuroEvolution for the value-based and temp-based problem. We use the same settings for the PGPE algorithm as before, we identify the optimal values for $\lambda_1$ and $\lambda_2$ (Eq. 5) based on a broad-sacle grid s.t. $\lambda_1 = 10\lambda_2$ or $\lambda_1 = 100\lambda_2$. The optimal parameters are reported in Appendix A.4. While

| Mask | Gradient Decent | | | | | NeuroEvolution | | | | |
|---|---|---|---|---|---|---|---|---|---|---|
| | F1 ↑ | Prec ↑ | Rec ↑ | I ↑ | S ↓ | F1 ↑ | Prec ↑ | Rec ↑ | I ↑ | S ↓ |
| MLP | 0.192 | 0.419 | 0.131 | 189 | **48.235** | 0.534 | 0.522 | **0.582** | **2387** | 2.050 |
| MLP-F | 0.250 | 0.796 | 0.167 | 268 | 79.126 | **0.622** | **1.000** | 0.452 | 1877 | **0.218** |
| MT | **0.978** | **1.000** | **0.957** | 335 | 238.789 | 0.219 | 0.147 | 0.436 | 449 | 134.712 |

Table 1: Comparison of different mask parameterizations and whether they are learned using gradient descent or NeuroEvolution. Performance is compared on the Rare Feature Dataset using differentiable perturbations as proposed in [4], averaged across 10 runs. The mask tensor (MT) has 5,000 entries (50 features × 100 time points). The hidden dimensions of MLP and MLP-F are scaled to approx. the same number of parameters, resulting in 4,914 and 4,898 parameters, respectively.

all three perturbations perform similarly for the value-based setting, Del, unsurprisingly, clearly outperforms FMA and GB for the temp-based setting, being the only perturbation to detect saliency that is solely based on $t$. Similarly, in Table 3, we observed comparable performance for perturbation FMA and Del for the value-based setting, yet a complete failure to detect saliency for perturbations FMA and GB in the temp-based setting. We generally observe lower F1 scores for the Rare Time setting than the Rare Feature Problem, indicating slightly higher difficulty.

| Perturbation | Value-based | | | | | Temp-based | | | | |
|---|---|---|---|---|---|---|---|---|---|---|
| | F1 ↑ | Prec ↑ | Rec ↑ | I ↑ | S ↓ | F1 | ↑ Prec ↑ | Rec ↑ | I ↑ | S ↓ |
| GB | 0.556 | 0.973 | 0.430 | 1786 | **0.004** | 0.000 | 0.000 | 0.000 | 0 | 0.000 |
| FMA | 0.638 | 0.774 | 0.557 | 2311 | 0.066 | 0.000 | 0.000 | 0.000 | 0 | 0.000 |
| Del (ours) | **0.891** | **0.967** | **0.851** | **3533** | **0.004** | **0.982** | **0.967** | **1.000** | **4152** | **0.004** |

Table 2: Method comparison on the Rare Feature Dataset across 10 runs using MLP-F masks. Examples of fitted masks are included in Appendix A.7.

In Table 4, we compare the explanations of MLP-F masks for the real-world sepsis prediction task. We trained an NCDE [8] and mtan [28] model for the sepsis task proposed in [8]. Both models which achieves a binary AUC of roughly 0.90 on a held-out test set (the same 20% split as per [8]). We compare the change in log-odds when perturbing $\approx 10\%$ of the observed data points that are indicated as important by the respective MLP-F mask. We train masks using an adapted version of Eq. 5, where we penalize not the total area covered by the mask but the deviation from the 10% of observed data (see Appendix A.5 for details). Since returned masks usually cover a larger area than 10%, we randomly sample them down to the desired 10%. We again start a grid search for different optimal penalty weights ($\lambda_1$ and $\lambda_2$ in Eq. 5), starting with very high values of $\lambda_1$ (we set $\lambda_2 = 0.1\lambda_1$ or $\lambda_2 = 0.01\lambda_1$). We iteratively reduce $\lambda_1$ and $\lambda_2$ by a factor of 10 until the resulting mask perturbs at least 10% of the observed data. Selected hyperparameters are outlined in Appendix A.8. An ablation study shows the effect of different values for $\lambda_1$ and $\lambda_2$ for the sepsis prediction task using the mtan model in Appendix A.6. Since evaluations of the NCDE model require calculating spline coefficients for every new perturbation, explaining NCDE models is very slow. The mtan model operates on 'raw' data without requiring additional calculations, and evaluations are much faster. When explaining a prediction made by either model, we observed a difference in speed of around 70x. For the NCDE model, we train the PGPE optimizer for only 200 iterations with a population size of 50, which brings a single run down to roughly 2 hours using an H200 GPU. The same process takes less than 2 minutes for the mtan model. We discuss computational speed in Appendix A.3 and A.8.

For the NCDE model, we explain the predictions for 50 participants of the test set (25 who were predicted to become septic and 25 who were predicted not to become septic). For the mtan model, we explain a total of 200 predictions for the 100 participants with the highest and lowest probability of becoming septic, respectively. We find that Del clearly outperforms FMA and GB for patients who are predicted to develop sepsis. Generally, all perturbations perform much better explaining Pred: Sepsis cases, i.e., where the model predicts sepsis. For cases that are predicted to not develop sepsis, value altering perturbations outperform Del. For predictions made using the mtan model, we investigate the explanations made using Contimask more closely in Section 5.1.

| Perturbation | Value-based | | | | | Temp-based | | | | |
|---|---|---|---|---|---|---|---|---|---|---|
| | F1 ↑ | Prec ↑ | Rec ↑ | I ↑ | S ↓ | F1 | ↑ Prec ↑ | Rec ↑ | I ↑ | S ↓ |
| GB | 0.336 | 0.794 | 0.283 | 396 | 0.003 | 0.000 | 0.000 | 0.000 | 0 | 0.000 |
| FMA | **0.747** | 0.997 | **0.600** | **840** | **0.001** | 0.000 | 0.000 | 0.000 | 0 | 0.000 |
| Del (ours) | 0.692 | **1.000** | 0.531 | 744 | **0.001** | **0.248** | **1.000** | **0.147** | **206** | 0.001 |

Table 3: Method comparison on the Rare Time Dataset across 10 runs using `MLP-F` masks. Examples of fitted masks are included in Appendix A.7.

| | mTAN | | | | NCDE | | | |
|---|---|---|---|---|---|---|---|---|
| | Pred: Sepsis | | Pred: No Sepsis | | Pred: Sepsis | | Pred: No Sepsis | |
| Perturbation | Del ↑ | Imp ↑ | Del ↑ | Imp ↑ | Del ↑ | Imp ↑ | Del ↑ | Imp ↑ |
| GB | 4.41 | **10.45** | 6.29 | **11.80** | 0.25 | 0.26 | 4.01 | 4.05 |
| FMA | 4.16 | 3.41 | **7.26** | 7.19 | 0.26 | 0.23 | **5.15** | **5.16** |
| Del (ours) | **7.88** | 2.26 | 4.57 | 5.90 | **4.97** | **0.21** | 2.72 | 3.94 |

Table 4: Comparison of perturbations using **Contimask** for Sepsis prediction from ICU data [8] using an MLP-F mask. We train an mTAN [28] and NCDE [8] model for the sepsis prediction task and explain predictions on the left-out test set. For mTAN, we investigate the 100 cases with the highest and lowest predicted probability of developing sepsis. For NCDE, we only investigate 25 cases each.

## 5.1 Case Study

For the Sepsis prediction task using the mtan model, we further investigated the results of fitted masks. We display 4 examples in Figure 2. Table 5 showcases the 6 features that were removed proportionally most often for a participant that was predicted to develop sepsis by the `mtan` model. Interestingly, these are mainly static features (age, height, ICUType), or features that are up to 52% more likely to have been recorded for a patient that later becomes septic (Imbalance). We include a full table in Appendix A.9 Table 13. Retention indicates how often a feature was

Table 5: Retention and sepsis-related imbalance for the 6 features removed most often by `Del`-perturbation for `mtan` sepsis predictions

| Feature | Retention (%) | Imbalance |
|---|---|---|
| Age | 79.12 | 1.00 |
| Height | 79.82 | 1.00 |
| AST | 80.63 | 1.52 |
| ICUType | 81.96 | 1.00 |
| ALP | 82.04 | 1.44 |
| Cholesterol | 83.00 | 1.43 |

retained after applying the learned `Deletion` mask. Since not all signals are recorded for all patients, there is the possibility of structural differences in the data between patients who develop sepsis and patients who do not. Examples of this might include blood markers that are only relevant to monitor if the treating physician believes that the patient is at a high risk of developing sepsis. The value in the Imbalance column indicates how much more likely a patient who later developed sepsis is to have any observation of that specific feature, highlighting the potential of bias related to this particular feature. A value greater than 1 indicates that this feature is recorded more often for patients who later become septic. Troponin I, a bloodmarker often used to test for sepsis, has the highest Imbalance with a value of 2.6. AST, ALP, and Cholesterol are among the next most imbalanced features, and are indicated to strongly influence predictions by the `mtan` model. This indicates that the `mtan` is highly influenced by decisions of treating physicians about whether to monitor certain features, which might be highly indicative of whether a patient is at risk of becoming septic. If this is indeed the case, such a model would be of little use in practice as it simply confirms the decision of a treating physician. The capability of irregular time series models to incorporate the structure of the data into the prediction can therefore also become an issue and introduce bias.

## 6 Discussion

We show that to explain irregular time series models, existing techniques for regular time series are only partially sufficient. We propose the **Contimask** framework, where we learn masks parameterized as small feed-forward networks with Fourier feature transformations using NeuroEvolution. Within

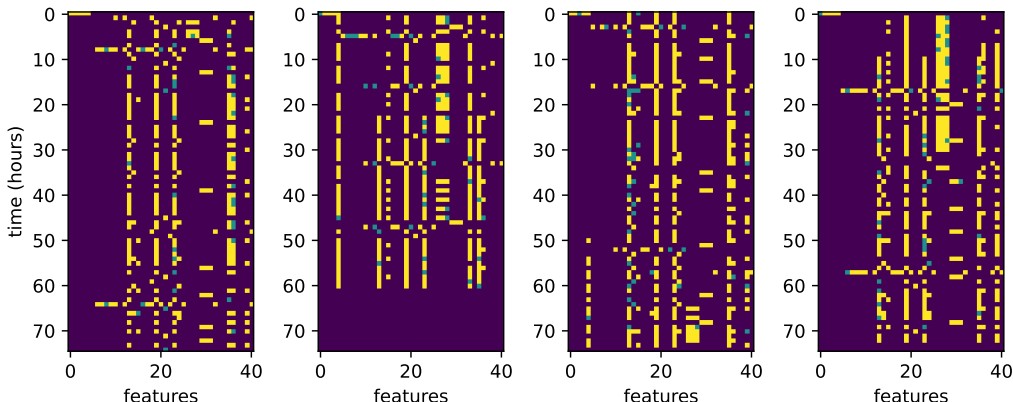

Figure 2: Example masks fitted to 4 participants predicted to develop sepsis. Observed features that are not affected by the perturbation are marked in yellow. Green dots are removed by the fitted mask.

this framework, we are able to uncover saliency that is based on the structure of the data, such as time intensity informative missingness, via our novel `Deletion` perturbation, where we simulate that data had not been observed. While our `Deletion` perturbation is non-differentiable (as any perturbation that alters the structure of the observed data), it clearly outperforms differentiable perturbations on toy datasets and sepsis predictions on real-world data. In particular, existing techniques for regular time series fail if the saliency is not dependent on the value of observed data, which Contimask using `Deletion` is able to detect. As indicated by the strong performance of Contimask using the `Deletion` perturbation when explaining a model for sepsis prediction, missingness and the structure of data are important to explain irregular time series models operating on hospital records. We find particular evidence of irregular time series models picking up on structural differences in the data introduced by decision of the treating physicians, which might introduce dangerous biases.

The applicability of Contimask is limited based on the model family that is being explained: while some architectures (e.g., `mtan`) are much quicker to explain, explaining `NCDE` models is particularly challenging given the repeated fitting of cubic splines. Further, Contimask is limited in its ability to incorporate learned perturbations similar to [5, 13], where more fine-tuned perturbations improve explanations, since NeuroEvolution does not scale as well as gradient descent. In practice, we have found that the slower speed of NeuroEvolution makes it more difficult to obtain stable results. Thus, improved NeuroEvolution strategies might lead to improvements in speed and also accuracy.

We thereby conclude that the development of novel post-hoc explanation tools for irregular time series is crucial to foster more broad-scale application of irregular time series in critical domains such as healthcare, where data is not missing at random. In such scenarios, it is critical to assess if the model is indeed useful in practice or has picked up biases that might also arise from informative missingness. Existing techniques for regular time series are not sufficient, and perturbations truly imitating irregular time series are needed for better model explanations. However, one of two advancements is necessary to increase the applicability of such explainability methods: Either, 1) advancements on *differentiable* perturbations that uncover similar patterns such as our introduced `Deletion` perturbation, or 2) advancements of existing NeuroEvolution methods to speed up the learning of masks such as the ones used in Contimask are needed. Since the latest generative models for irregular time series are not capable of altering the timing and structure (i.e., missingness) of data [19], generating more diverse counterfactual examples in a differentiable manner still remains a very difficult problem. Tailoring NeuroEvolution techniques to train masks for 2D problems more efficiently might be a more promising route for future work.

Further work is needed to combine value-altering and structure-altering perturbations into a single framework. Existing techniques for regular time series might be able to pick up saliency that Contimask using the `Deletion` perturbation is not, as also indicated by our results for sepsis-negative cases where value-altering perturbations performed best.

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

# A  Appendix

## A.1  Examples Of Perturbation In Continuous Time

Figure 3 shows examples of the three perturbations applied to the same single-channel irregular time series. All perturbations are applied to the same area of that time series.

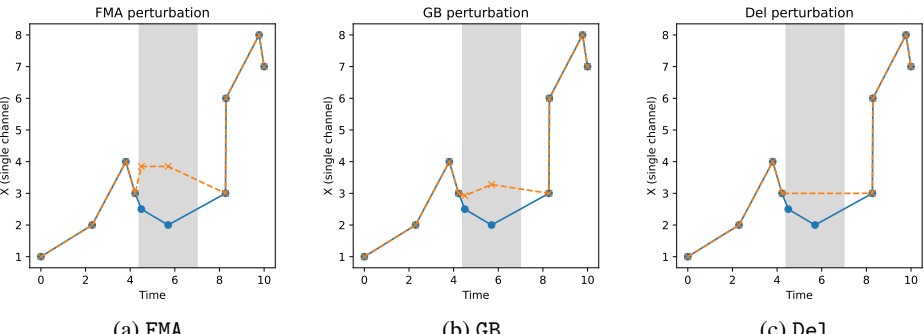

(a) `FMA`          (b) `GB`          (c) `Del`

Figure 3: Examples of the three perturbations `FMA`, `GB`, and `Del` applied to same single channel irregular time series in the gray-shaded area. `FMA` insert the Moving Average, `GB` applies a Gaussian Blur, and `Del` deletes the points that lie inside the gray-shaded area.

## A.2  Simulating The Deletion Of Data In A Differentiable Manner

Without knowing how exactly a model processes the data it receives as input, it is not possible to simulate that data had not been observed in a way that is differentiable with respect to the mask $m$, where $m$ highlights which data points are being deleted. In such a case, $m$ needs to be converted to a hard mask. Essentially, at some point during the process, $m$ needs to function as an index. This means it has to be converted to an integer or boolean data type, which breaks the automatic differentiation graph in frameworks such as PyTorch.

For `NCDE` models, for instance, the way in which data is processed internally allows to (partially) simulate that data had not been observed. `NCDE` models model irregular data using differential equations. Via integration (i.e., the numerical ODE solver of one's choice), the hidden states of the model are derived in continuous time. Given the nature of differential equations, the hidden states will therefore only update if the data changes. One can therefore delete the data observed at a certain timepoint by constructing a forward (or indeed backward) fill that overwrites said timepoint to duplicate one of the neighboring timepoints. Since this only involves multiplication and addition, we are still able to backpropagate w.r.t. $m$. However, note that this includes overwriting the time intensity channels, and also the entry in $t$. Thereby, one can alter that no new data had been observed at the chosen timepoint. This means simulating that no data had been observed at all at that timepoint—across all channels.

Going one step further, it is possible to simulate the deletion of data points in a way that is differentiable with respect to the deletion mask $m$ for models such as `mtan` if one has access to model internals. Since `mtan` relies on attention modules to deal with missing data, it is possible to manipulate the attention mask to simulate that some further datapoints had not been observed. To do so, one requires access to the attention module. `mtan` uses scaled dot-product attention for some reference points (set somewhat arbitrarily to project the irregular data into a fixed dimension) and the timings of observed data points. The points one is trying to 'delete' can then be subtracted from the resulting attention mask or the corresponding part is simply multiplied by zero during the dot-product operation. These points then no longer influence the state of the model at the reference points, and thereby no longer have influence on the model output. Since this operation does not require a hard mask, and only involves multiplication and subtraction, it allows to simulate that data had not been observed if one is willing to access model internals.

Please note that we have not tested the ideas described above in practice. One would need to verify that they behave exactly as intended and indeed produce gradients useful for learning. While the

above strategies for `mtan` and `NCDE` do not break gradients, it is not guaranteed that they indeed feedback gradients that enable stable learning. If one is using an objective similar to Eq. 5, the gradients related to the penalty terms will not be zero, but gradients related to the error in $f$ will likely oftentimes be zero which might still make learning difficult.

## A.3   NeuroEvolution: performance, parameter counts, and runtime

In Table 6 we compare the runtime and performance when training `MLP-F` masks using different parameter counts. While performance varies based on parameter count, the time required for training remains relatively constant. All experiments were performed on a H200 GPU, where the used VRAM never exceeded 8 GB. In Table 6, the performance is averaged across the 10 runs, while the runtime is for all 10 experiments together. The results show that a much smaller mask with only 1906 parameters achieves similar performance to a mask close to 5000 parameters.

Training using NeuroEvolution takes 4–5 times longer than using gradient descent. As discussed in Section 5, we observe that masks parameterized as a Tensor of the same shape as $X$ and $d$ train in much fewer iterations using gradient descent, but require more iterations using NeuroEvolution. We've trained for 2,000 iterations using NeuroEvolution, and 32,000 iterations using gradient descent. Masks parameterized as `MLP-F` converge much better using NeuroEvolution but worse using gradient descent. We can only speculate about the reasons for this. Potentially, they are related to the assumptions algorithms such as PGPE make about the distributions of the parameters they are trying to optimize.

|  | Model Size | | | NeuroEvolution | | | | Gradient Descent | | | |
|---|---|---|---|---|---|---|---|---|---|---|---|
|  | Dim | L | Count | F1 | Prec | Rec | Time | F1 | Prec | Rec | Time |
| MLP-F | 32 | 34 | 4914 | 0.622 | 1.000 | 0.452 | 2:18 | 0.250 | 0.796 | 0.167 | 0:27 |
| MLP-F | 32 | 32 | 4786 | 0.690 | 0.961 | 0.556 | 2:18 | 0.170 | 0.372 | 0.115 | 0:26 |
| MLP-F | 32 | 24 | 4274 | 0.638 | 0.774 | 0.557 | 2:08 | 0.170 | 0.569 | 0.112 | 0:26 |
| MLP-F | 16 | 24 | 1906 | 0.670 | 0.897 | 0.560 | 2:17 | 0.218 | 0.560 | 0.143 | 0:26 |
| MLP-F | 16 | 12 | 1522 | 0.584 | 0.835 | 0.490 | 2:18 | 0.157 | 0.377 | 0.108 | 0:27 |
| MLP-F | 8 | 24 | 914 | 0.539 | 0.717 | 0.515 | 2:17 | 0.107 | 0.381 | 0.068 | 0:26 |
| MLP-F | 8 | 12 | 722 | 0.497 | 0.806 | 0.429 | 2:13 | 0.106 | 0.353 | 0.070 | 0:27 |

Table 6: Model size, performance, and runtime averaged across 10 runs for the value-based Rare Feature problem when training `MLP-F` masks using the `FMA` perturbation across 10 runs. The performance is averaged across 10 runs, the runtime (Time) is given for all 10 runs in total in (h:mm), i.e., training an `MLP-F` mask with 4914 parameters 10 times takes 2 hours and 18 minutes using NeuroEvolution and only 27 minutes using gradient descent.

## A.4   Regularizing Parameters

Table 7 and 8 display the chosen values for $\lambda_1$ and $\lambda_2$ for the Rare Feature and Rare Time problems for each of the two settings. We started with values for $\lambda_1$ that penalize the area covered by the mask too heavily and then decreased $\lambda_1$ until the mask returned reasonable results, where initially we restricted $\lambda_2$ such that $\lambda_1 = 10\lambda_2$ or $\lambda_1 = 100\lambda_2$. Once a value for $\lambda_1$ was set, we optimized further for $\lambda_2$ also allowing for smaller values of $\lambda_2$.

|  | Value-based | | Temp-based | |
|---|---|---|---|---|
| Perturbation | $\lambda_1$ | $\lambda_2$ | $\lambda_1$ | $\lambda_2$ |
| GB | 0.01 | 0.001 | 0.1 | 0.000001 |
| FMA | 0.01 | 0.001 | 0.1 | 0.000001 |
| Del | 0.1 | 0.001 | 0.1 | 0.000001 |

Table 7: Values for $\lambda_1$ and $\lambda_2$ for the Rare Feature Problem reported in Table 2.

| | Value-based | | Temp-based | |
|---|---|---|---|---|
| Perturbation | $\lambda_1$ | $\lambda_2$ | $\lambda_1$ | $\lambda_2$ |
| GB | 0.1 | 0.0001 | 0.00001 | 0.000001 |
| FMA | 0.1 | 0.0001 | 0.00001 | 0.000001 |
| Del | 0.1 | 0.0001 | 0.00001 | 0.000001 |

Table 8: Values for $\lambda_1$ and $\lambda_2$ for the Rare Time Problem reported in Table 3.

## A.5  Updated Objective Function For Real-world Sepsis Task

For the sepsis prediction task, we adapted Equation 5 such that the loss term involving $\lambda_1$ changes from:

$$\lambda_1 \sum_C \int_0^T (1 - m(u)_c) \, du \tag{9}$$

to:

$$\lambda_1 \left( TS - \sum_C \int_0^T (m(u)_c) \, du \right), \tag{10}$$

where TS is the target size, i.e. in our case $100\% - 10\% = 90\%$.

For the sepsis task, the goal is to maximize the divergence in prediction while the mask should only alter a previously defined target size $TS$, for example, 10% of all observed time points. The goal for the previous tasks on synthetic data was to keep the prediction unchanged while altering as many data points as possible.

The full objective for the sepsis task then becomes:

$$\min_m -\mathcal{L}(f(t_n, x_n, d_n), (\tilde{t}_n, \tilde{x}_n, \tilde{d}_n)) + \lambda_1 \left( TS - \sum_C \int_0^T (m(u)_c) \, du \right) + \lambda_2 \sum_C \int_0^T |m(u)'_c| \, du. \tag{11}$$

## A.6  Ablation Study for $\lambda_1$ and $\lambda_2$

We conducted an ablation study on the hyperparameters $\lambda_1$ and $\lambda_2$ for the Del, FMA, and GB attribution methods on the sepsis prediction task using mTAN. The results are reported for 10 cases predicted as sepsis-positive. Each cell contains two values: the first corresponds to the Del odds change, and the second to the Val odds change. Note that these results can be higher than those reported for the full set of 100 sepsis-positive cases, as the top 10 predictions are typically easier to explain—more confident predictions tend to exhibit more pronounced feature relevance.

Table 9: Ablation on $\lambda_1$ and $\lambda_2$ for Del. Each cell shows Del odds change | Val odds change.

| $\lambda_1$ | $\lambda_2 = 0$ | $\lambda_2 = \lambda_1/100$ | $\lambda_2 = \lambda_1/10$ |
|---|---|---|---|
| 0.01 | 6.24 | 7.06 | 10.34 | 7.94 | 5.95 | 8.27 |
| 0.1 | 9.13 | 8.44 | 9.91 | 7.81 | 6.51 | 6.82 |
| 0.5 | 10.75 | 8.83 | 9.05 | 6.89 | 8.09 | 7.42 |
| 1.0 | 12.92 | 8.42 | 13.64 | 7.19 | 10.99 | 8.48 |
| 10.0 | 11.24 | 6.81 | 8.50 | 7.55 | 8.69 | 7.30 |

## A.7  Examples of Fitted Masks

To visualize how metrics such as F1-Score, Precision, and Recall represent the quality of fitted masks, Figure 4 shows examples of MLP-F masks fitted based on GB, FMA, and Del perturbations using NeuroEvolution. The masks are fitted for the value-based Rare-Feature problem and correspond to what is displayed in Table 2 (100 time points and 50 features). Averaged across the 10 runs, the

Table 10: Ablation on $\lambda_1$ and $\lambda_2$ for FMA. Each cell shows Del odds change | Val odds change.

| $\lambda_1$ | $\lambda_2 = 0$ | $\lambda_2 = \lambda_1/100$ | $\lambda_2 = \lambda_1/10$ |
|---|---|---|---|
| 0.01 | 6.76 | 7.76 | 6.01 | 7.81 | 10.08 | 7.48 |
| 0.1 | 7.78 | 8.15 | 9.50 | 8.95 | 9.68 | 9.31 |
| 0.5 | 7.16 | 6.83 | 10.47 | 8.17 | 12.63 | 9.49 |
| 1.0 | 8.10 | 6.95 | 10.24 | 9.15 | 11.83 | 7.15 |
| 10.0 | 11.74 | 7.04 | 11.63 | 6.56 | 11.43 | 7.71 |

Table 11: Ablation on $\lambda_1$ and $\lambda_2$ for GB. Each cell shows Del odds change | Val odds change.

| $\lambda_1$ | $\lambda_2 = 0$ | $\lambda_2 = \lambda_1/100$ | $\lambda_2 = \lambda_1/10$ |
|---|---|---|---|
| 0.01 | 5.86 | 15.78 | 5.54 | 15.58 | 8.91 | 15.16 |
| 0.1 | 4.64 | 15.22 | 5.85 | 14.84 | 9.14 | 16.04 |
| 0.5 | 6.05 | 16.92 | 9.27 | 17.02 | 8.59 | 15.48 |
| 1.0 | 7.04 | 16.15 | 8.31 | 16.82 | 9.30 | 14.62 |
| 10.0 | 7.95 | 8.31 | 10.99 | 11.16 | 9.86 | 11.93 |

masks based on GB correspond to an F1 Score of 0.556, compared to 0.638 for masks based on FMA and 0.891 for masks based on Del. The ground truth saliency changes every run (i.e. changes row by row) but is the same for all perturbations (i.e., only one ground truth saliency per row).

## A.8   Details On The Sepsis Prediction Task

We train a model according to [8] for sepsis prediction on real-world hospital data. Kidger et al. [8] feed cubic spline coefficients to the model. Since the computation of these coefficients takes long if there are missing data in $X$, they compute and save the coefficients once and simply load them during iterative training and testing. Since we alter the underlying data using our perturbations, we have to recompute the cubic spline coefficients at every iteration, which is very costly. Computing the coefficients according to Kidger et al. [8] takes approximately 30 seconds for a single triplet $(t, X, d)$. While this is largely parallelized, this significantly increases the time needed to explain a single prediction by the model. As per Table 6, 10 runs to using NeuroEvolution take around 2 hours and 18 minutes. This is roughly the time it takes to conduct one run for the real-world sepsis prediction task. Note that we have already decreased the iterations for PGPE from 2000 to 200. While the reduction to 200 iterations cuts computation time, it leads to poorer fitted saliency maps. We have not been able to quantify this drop in performance.

For other models such as mtan, where the raw data is fed into the models, perturbing data is much more straight forward and such experiments will be much closer to the Rare Feature and Rare Time examples in terms of required compute.

Table 12 shows the $\lambda_1$ and $\lambda_2$ values for the real-world sepsis prediction task using the NCDE model. Similar to before, we started with too large values of $\lambda_1$ and set $\lambda_1 = 10\lambda_2$ or $\lambda_1 = 100\lambda_2$. We then iteratively reduced $\lambda_1$ until we reached reasonable performance and then further finetuned $\lambda_2$. Generally, we did not consider alternative values for $\lambda_1$ and $\lambda_2$ other than multiplies of $10^n$ for some positive or negative integer $n$.

For the mtan model, we simply set $\lambda_1 = 1$, and $\lambda_2 = 0$ (ablation study above).

| Perturbation | Pred: No Sepsis | | Pred: Sepsis | |
|---|---|---|---|---|
| | $\lambda_1$ | $\lambda_2$ | $\lambda_1$ | $\lambda_2$ |
| GB | 1 | 0.01 | 1 | 0.01 |
| FMA | 0.00001 | 0.000001 | 0.00001 | 0.000001 |
| Del | 1 | 0.01 | 10 | 0.1 |

Table 12: Regularization parameters for the real-world sepsis prediction task using the NCDE model.

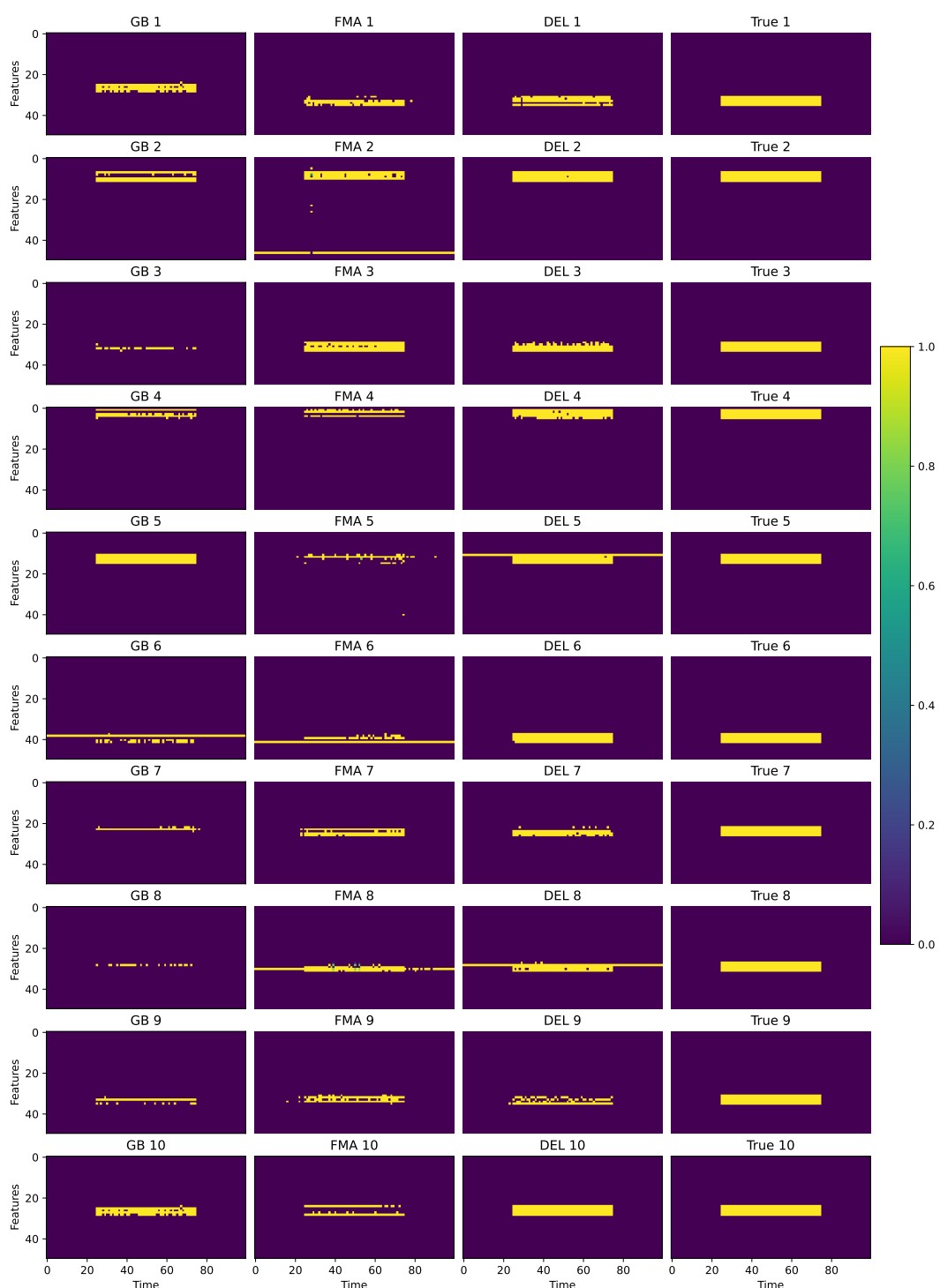

Figure 4: Examples of fitted masks using `GB`, `FMA`, and `Del` for the value-based Rare Feature problem (`MLP-F` masks trained using NeuroEvolution). The masks correspond to the results is displayed in Table 2.

## A.9   Mask Evaluation For Sepsis Prediction Task Using mtan Model

We evaluated for every feature of the data how often it was removed during the real-world sepsis task based on the `mtan` model. If a feature was removed very often, i.e., retention is low, it is estimated

to have a large effect on the prediction. In Table 13, we ordered all features by retention ratio, i.e., the most important features are listed first. It is a more extensive version of Table 5. In the Sepsis Imbalance column, we've calculated how much more often this feature was present for a patient who later becomes septic compared to a patient who does not become septic. High values indicate that the shear presence of a particular feature might be a strong indicator of a patient later developing sepsis. This corresponds to decisions of the treating physicians who might decide to only collect certain information if a patient is at a high risk of developing sepsis.

| Feature | Retention Ratio | Sepsis Imbalance | Description |
| --- | --- | --- | --- |
| Age | 0.7912 | 1.0000 | Patient age (y) |
| Height | 0.7982 | 1.0000 | Height (cm) |
| AST | 0.8063 | 1.5256 | Aspartate transaminase |
| ICUType | 0.8196 | 1.0000 | ICU unit type |
| ALP | 0.8204 | 1.4433 | Alkaline phosphatase |
| Cholesterol | 0.8300 | 1.4276 | Serum cholesterol |
| RespRate | 0.8360 | 0.5092 | Respiratory rate (bpm) |
| Creatinine | 0.8422 | 1.0123 | Serum creatinine |
| K | 0.8589 | 1.0100 | Potassium (mmol/L) |
| MechVent | 0.8597 | 1.1880 | Mechanical ventilation |
| Na | 0.8610 | 1.0100 | Sodium (mmol/L) |
| pH | 0.8656 | 1.1483 | Arterial pH |
| NISysABP | 0.8741 | 1.0183 | Non-invasive systolic BP |
| Gender | 0.8790 | 1.0000 | Sex (F=0, M=1) |
| MAP | 0.8793 | 1.0734 | Mean arterial pressure |
| DiasABP | 0.8846 | 1.0689 | Diastolic BP |
| Weight | 0.8852 | 1.0000 | Weight (kg) |
| PaO2 | 0.8902 | 1.1642 | Arterial oxygen (mmHg) |
| SysABP | 0.8935 | 1.0689 | Systolic BP |
| Temp | 0.8946 | 1.0102 | Temperature (°C) |
| Platelets | 0.8956 | 1.0138 | Platelet count |
| TroponinT | 0.8998 | 1.4946 | Cardiac troponin T |
| HR | 0.9038 | 1.0102 | Heart rate (bpm) |
| GCS | 0.9057 | 1.0102 | Glasgow Coma Scale |
| Urine | 0.9058 | 0.9904 | Urine output (mL) |
| Lactate | 0.9093 | 1.3661 | Blood lactate |
| ALT | 0.9104 | 1.5256 | Alanine transaminase |
| Glucose | 0.9114 | 1.0191 | Blood glucose |
| HCT | 0.9174 | 1.0123 | Hematocrit (%) |
| TroponinI | 0.9186 | 2.6003 | Cardiac troponin I |
| SaO2 | 0.9190 | 1.0734 | $O_2$ saturation (%) |
| NIDiasABP | 0.9236 | 1.0200 | Non-invasive diastolic BP |
| FiO2 | 0.9239 | 1.2318 | Inspired $O_2$ fraction (%) |
| NIMAP | 0.9337 | 1.0200 | Non-invasive MAP |
| HCO3 | 0.9370 | 1.0115 | Bicarbonate (mmol/L) |
| BUN | 0.9421 | 1.0123 | Blood urea nitrogen |
| Bilirubin | 0.9458 | 1.4518 | Total bilirubin |
| PaCO2 | 0.9475 | 1.1642 | Arterial $CO_2$ (mmHg) |
| WBC | 0.9551 | 1.0168 | White blood cells |
| Albumin | 0.9648 | 1.3987 | Serum albumin |
| Mg | 0.9721 | 1.0191 | Magnesium (mmol/L) |

Table 13: Features included in the real-world sepsis prediction task ordered by their estimation importance (most important features first) including the retention ratio, an indicator of structural bias (Imbalance) of this feature, and a short description.

