# OpenReview forum: "Contimask: Explaining Irregular Time Series via Perturbations in Continuous Time"
_NeurIPS.cc/2025/Conference — NeurIPS 2025 poster_

### Official Review · Reviewer_yKqd · 2025-07-02

**Clarity:** 3
**Significance:** 2
**Originality:** 4
**Rating:** 5
**Confidence:** 3

**Summary:**

The authors provide a novel interpretability method, Contimask, for irregular time-series data which is sampled irregularly and can having missing values. The algorithm works by perturbing the data with a perturbation function f to find which changes in input have the largest effect on a model's prediction. While machine learning currently focuses on gradident descent, this assumes access to a differentiable function f, which the authors don't assume access to so they leverage a genetic algorithm. This allows them to define novel perturbation strategies that are not differentiable but effective. They test their algorithm on 3 datasets with 2 baselines.

**Questions:**

1. Can you provide a qualitative display of the interpretability enabled by your method? This seems important for a method in the interpretability literature.
2. Can you provide more baselines?

**Ethical Concerns:**

["NO or VERY MINOR ethics concerns only"]

**Final Justification:**

They have added more baselines and promised to add the kind of visualization I think is important for this kind of method.

**Limitations:**

Yes.

**Quality:**

3

**Strengths And Weaknesses:**

quality. The paper is clear and high-quality.

clarity: the paper is clear. more could be see about how much computation and time is needed for running experiments given that speed is a limitation of the method.

significance: the paper could do more comparisons of baselines. 2 does not seem sufficient. I appreciate the ablations on the method between gradient descent and neuroevolution approaches.

originality: this is the first method for irregular time-series.

---

> ### Author Rebuttal · Authors · 2025-07-31
>
> We'd like to thank the reviewer for their time and thoughtful comments, which we have incorporated in our revision to refine and strengthen our paper. We were delighted to hear they found our paper 'clear and high-quality'. In the below, we hope to answer all raised questions.
>
> **Q:** Qualitative display of interpretability
>
> **R:** We thank the reviewer for this suggestion. While we previously included visualizations of fitted masks on synthetic data, we agree that mask visualizations for the sepsis prediction task would also be of great value.
>
> As several reviewers raised concerns about computational speed, we investigated potential bottlenecks in our code more thoroughly. We identified that the main bottleneck for the sepsis task lies in the computation of cubic spline coefficients as part of the Neural CDE (NCDE) model. While we noted this as a potential issue in the original manuscript, we had underestimated its impact. To better understand this, we trained a multi-time attention network (mTAN) on the same sepsis task. Since mTAN processes raw time series directly, model interactions are significantly faster. This resulted in a 70× speed-up, reducing a process that previously required ~2 hours with the NCDE model to under 2 minutes with mTAN. We have run a much more extensive evaluation using the mTAN model and conducted also a qualitative analsys.
>
> In our qualitative analysis of the mTAn-model sepsis task, we observed particularly interesting cases where entire sensor streams (e.g., blood markers highly predictive of sepsis such as ALT, AST, Troponin) were removed by Contimask leading to drastically lower probabilities of sepsis. Interestingly, sepsis-positive patients are up to 2.6 times more likely to have data for the aforementioned sensor streams than sepsis-negative patients. This suggests that clinicians tend to adapt monitoring setups based on patient condition and patients at higher risk of sepsis are monitored more closely.
>
> ContiMask often removes these sensor streams for sepsis-positive cases, leading to dramatic changes in the model’s predictions. This reveals that the trained mTAN and NCDE model may rely heavily on patterns related to clinician monitoring decisions, rather than intrinsic patient data, which limits its practical utility. The model therefore provides little additional value for a treating clinician since it uses a decision made by a clinician as its main discriminator. The ability to spot this is something unique to Contimask and the Deletion perturbation.
>
> For sepsis-negative cases, data removal is less effective in altering predictions because these patients are typically monitored less extensively. In such cases, value-altering perturbations are more effective than the Deletion perturbation at changing the model’s predictions.
>
> We will include these visualizations and a more detailed discussion in the final version of the manuscript.
>
>
> **Q:** baselines
>
> **R:** We extended our experiments to also explain a multi-time attention network (mTAN) in addition to the Neural CDE. TimeX would most liekly struggle to explain the NCDE model given the CDE solve. While TimeX could, in principle, explain models such as mTAN, we were unable to reproduce its results using the provided repository. The original implementation relies on a graph-neural-network–based model for irregular time series, but links for data download are broken, and it is unclear how to train the model or integrate a model like mTAN with the explainer.
>
> Moreover, TimeX requires access to model internals and inspects the latent space, whereas Contimask only queries the model, making a direct comparison less fair. Nonetheless, this would be an interesting direction for future work. At present, no immediately applicable baseline exists that performs post-hoc instance explanation for time series models without the need to access model internals.

---

> > ### Comment · Reviewer_yKqd · 2025-08-05
> > **response**
> >
> > Thanks for the thorough response. Thanks for adding baselines and adding visualization of interpretability (it's a shame that you can't change the manuscript, so we just have to trust you). But I'm okay with that if the AC is.

---

> > > ### Author Response · Authors · 2025-08-06
> > >
> > > Thank you for your time and valuable feedback. If there are any remaining concerns we can address before the 8th, please don’t hesitate to let us know.
> > >
> > > Based on the Program Chairs’ instructions, we understood that we were not permitted to share images, links to any materials, or an updated version of our manuscript. If there is an alternative way for us to provide the visualizations, we would be happy to do so.

---

### Official Review · Reviewer_AND9 · 2025-07-03

**Clarity:** 2
**Significance:** 3
**Originality:** 2
**Rating:** 4
**Confidence:** 4

**Summary:**

This paper addresses the challenge of explaining black-box models for irregular time series (ITS) data, which are prevalent in domains like healthcare, where observations occur at non-uniform intervals with high missingness rates. The authors identify limitations in existing saliency methods designed for regular time series, which fail to account for the timing/structure of observations in ITS. They propose ​Contimask, a framework that:

- Translates perturbation techniques (e.g., Gaussian blur, moving average) to the continuous-time domain of ITS.
- Introduces a novel ​Deletion perturbation​ (Del) that simulates unobserved data points via Bernoulli sampling, enabling the detection of saliency related to missingness and time intensity.
- Utilizes ​NeuroEvolution​ (PGPE algorithm) to optimize non-differentiable perturbations, parameterizing masks as MLPs with Fourier features (MLP-F) for efficiency.

**Questions:**

Q1. Contimask was tested on only two sepsis cases due to computational costs. How would it perform on larger datasets (e.g., 100+ patients)? Can the authors provide evidence of scalability (e.g., parallelization, approximations) or runtime metrics (e.g., avg. time per mask)?

Q2. Del removes data points via Bernoulli sampling, potentially creating unrealistic "gaps" if adjacent timestamps are masked. How does this affect fidelity in clinical contexts (e.g., where missingness patterns are structured)?

Q3. λ₁ and λ₂ in Eq. 5 critically affect mask sparsity/smoothness. How were they optimized for sepsis tasks? Can the authors show ablation studies on synthetic data?

**Ethical Concerns:**

["NO or VERY MINOR ethics concerns only"]

**Final Justification:**

Most of my concerns have been well addressed by the author's thorough responses. Therefore, I have improved my score.

**Limitations:**

yes

**Quality:**

2

**Strengths And Weaknesses:**

### Quality:​​

- Strengths: The problem formulation is well-motivated, especially for healthcare applications. The integration of NeuroEvolution to handle non-differentiable perturbations is innovative. Experiments on synthetic data validate Del’s ability to detect time-based saliency (Table 2-3).

- ​Weaknesses: Experimental validation is ​severely limited. The sepsis task evaluates only two cases due to computational constraints (Sec. 4), undermining statistical significance. Results in Table 4 show marginal improvements, but lack rigorous comparison across multiple instances or baselines (e.g., TimeX++ for ITS). The claim of "significant improvements" is overstated given the tiny sample size.

### Clarity:​​

- ​Strengths: The paper is generally well-structured, with a clear background on ITS models (Sec. 2.2) and method details (Sec. 3).

- Weaknesses: Critical details are ambiguous. For example: (i) How hyperparameters (λ₁, λ₂) were tuned for sepsis tasks is not specified (only mentioned as "adapted version of Eq. 5"); (ii) The computational overhead of NeuroEvolution (vs. gradient-based methods) is discussed but not quantified.

### Significance:​​

- Strengths: The focus on ITS explainability fills a genuine gap, especially for clinical applications. The Deletion perturbation is a novel concept for capturing missingness importance.

- ​Weaknesses: Practical utility is ​questionable. Contimask’s reliance on NeuroEvolution makes it prohibitively slow (2 hours per run for sepsis cases), limiting real-world adoption. No comparison with SOTA ITS explainers (e.g., TimeX) weakens impact claims.

### Originality:​​

- Strengths: The extension of perturbations to continuous time and the Del operator are non-trivial contributions. MLP-F masks offer parameter efficiency (Table 1).

- ​Weaknesses: NeuroEvolution for explainability is not new (e.g., evolution strategies in RL). The translation of FMA/GB perturbations to ITS (Sec. 3) feels incremental, lacking fundamental innovation beyond prior work (DynaMask).

---

> ### Author Rebuttal · Authors · 2025-07-31
>
> We'd like to thank the reviewer for their time and thoughtful comments, which we have incorporated in our revision to refine and strengthen our paper. We were delighted to hear they found our paper 'well-motivated' and 'innovative' (AND9). In the below, we hope to answer all raised questions.
>
> **Q:** Quantification of computational overhead of NeuroEvolution
>
> **R:** We quantified the computational overhead in terms of runtime in Table 5 in the Supplementary Material. In a differentiable set-up based on the FMA perturbation, we found NeuroEvolution to take around 4 times longer than gradient-based methods as implemented for instance in DynaMask across 10 test runs. Interestingly, gradient-based optimization not always converge in these experiements, however.
>
> **Q:** Computational cost of Contimask for sepsis task and evluation on larger dataset
>
> **R:** As several reviewers raised concerns about computational speed, we investigated potential bottlenecks in our code more thoroughly. We identified that the main bottleneck for the sepsis task lies in the design of the Neural CDE (NCDE) model. This model requires cubic spline coefficients as input rather than the raw time series. Consequently, at every model interaction, cubic spline coefficients must be recalculated, which is computationally expensive—particularly in the presence of missing data, which limits parallelization.
>
> While we noted this as a potential issue in the original manuscript, we had underestimated its impact. To better understand this, we trained a multi-time attention network (mTAN) on the same sepsis task. Since mTAN processes raw time series directly, model interactions are significantly faster. This resulted in a 70× speed-up, reducing a process that previously required ~2 hours with the NCDE model to under 2 minutes with mTAN. Importantly, we only changed the model we are explaining. Hence, the slow speeds were related to the NCDE model, not Contimask.
>
> Below, we report the average change in model output for 100 sepsis-positive and 100 sepsis-negative cases predicted by mTAN, rather than the NCDE model. While the Del perturbation outperforms FMA and GB for sepsis-positive cases, GB performs best for sepsis-negative cases. We will include this table in the final version of the manuscript.
> |       | **Sepsis**      |                | **No-sepsis**    |                |
> |-------|-----------------|----------------|------------------|----------------|
> |       | Del Odds Change | Imp Odds Change| Del Odds Change  | Imp Odds Change|
> | GB    | 4.41            | 10.45          | 6.29             | 11.80          |
> | FMA   | 4.16            | 3.41           | 7.26             | 7.19           |
> | Del   | 7.88            | 2.26           | 4.57             | 5.90           |
>
> Further, we have also extended the sepsis task based on the NCDE model to 25 cases each: 25 cases that were intially predicted to develop sepsis, and 25 cases that were intially not predicted to develop sepsis. The below table confirms the results depicted in the original paper and observed also in the Table above. The Del perturbation works best at explaining cases predicted to be sepsis-positive, while FMA/GB work better for cases predicted not to develop sepsis. All models were trained with λ1=1, and λ2=0 (ablation study below).
>
> |       | **Sepsis**      |                | **No-sepsis**    |                |
> |-------|-----------------|----------------|------------------|----------------|
> |       | Del Odds Change | Imp Odds Change| Del Odds Change  | Imp Odds Change|
> | GB    | 0.25            | 0.26           | 4.01             | 4.05           |
> | FMA   | 0.26            | 0.23           | 5.15             | 5.16           |
> | Del   | 4.97            | 0.21           | 2.72             | 3.94           |
>
>
> Why the Del perurbation works better for cases predicted to be sepsis-positive relates actually relates to the question raised below. Please see our response below.
>
> **Q:** Gaps created by Bernoulli sampling and fidelity in clinical contexts (where missingness patterns might be structured)
>
> **R**: To remove patches of data, i.e. to create larger gaps, and potentially remove an entire sensor stream was in fact one of our motivations to use Bernoulli sampling in the first place. Related to the review by yKqd03, we investiagted returned masks, which showcased the strength of Contimask and the Deletion perturbation.
>
> In our analysis of the mTAN model, we observed for many sepsis-positive cases that Contimasks using the Deletion perturbation would remove entire sensor streams (mainly blood markers highly predictive of sepsis such as ALT, AST, Troponin), which led to drastically lower probabilities of sepsis (accroding to the models). Interestingly, sepsis-positive patients are up to 2.6 times more likely to have data for the aforementioned sensor streams than sepsis-negative patients. This suggests that clinicians tend to adapt monitoring setups based on patient condition and patients at higher risk of sepsis are monitored more closely.
>
> Using the Del perturbation, Contimask reveals that both mTAN and NCDE are both heavily influenced by data being observed in these channels and use this as a main discriminator of whether a patient will become septic. Of course, whether these channels are observed is decided by the treating physicians who decide which information to monitor. Thereby, the models do not actually provide the clincians with any meaningful information but oftentimes just use the clinicians judgement as a discriminator.
>
> For cases prediced as septic-negative, removing data does not increase the predicted probability of developing sepsis as strongly and value altering perturbations (i.e., FMA and GB) produce better results.
>
> We believe these findings highlight Contimask's unique capabilites to uncover the reliance of irregular time series on whether data was observed at all, especially if using the Del perturbation. In clinical settings, this can help to reveal that a trained model does not actually uncovered meaningful patterns to help the treating physicians.
>
> Removing entire patches of data is crucial for spotting this.
>
> **Q:** Tunig of λ₁ and λ₂ in Eq. 5 for the sepsis tasks
>
> **R:** We thank the reviewer for raising this. We adapted equation 5 such that the loss term involving λ1 changes from:
>
> $\lambda_1 \sum_C \int_0^T(1-m(u)_c) du$
>
> to:
>
> $\lambda_1 (TS - \sum_C \int_0^T( m(u)_c ) du)$.
>
> For the sepsis task, the goal is to maximize the divergence in prediction while the mask should only alter a previously defined target size $TS$, say 10% of all observed timepoints. The goal for the previous tasks on synthetic data was to keep the predition unchanged while altering as many datapoints as possible.
>
> The full obective for the sepsis task then becomes:
> $\min_m -\mathcal{L}(f(t_n, x_n, d_n), (\tilde{t}_n, \tilde{x}_n, \tilde{d}_n)) + \lambda_1 (TS - \sum_C \int_0^T( m(u)_c ) du) + \lambda_2 \sum_C \int_0^T \vert m(u)'_c \vert du$.
> We will add this in the final version of the manuscript. Intitially, we removed it to save space.
>
> Tuning 𝜆1 and 𝜆2 is generally straightforward. If no target mask size is enforced, setting λ1 too high can dominate the model output and collapse the mask. Even with a target size, overly large values may still slow convergence or reduce performance. In the sepsis task, where model outputs ranged from −15 to 15, λ1=1 produced reasonable results, so we conducted a focused search around this value, with 𝜆2 set to 0, λ/10, or λ1/100.
>
> The tables below show an ablation study for 10 cases predicted to bevome septic by mTAN. Each cell reports Del odds change and Imp odds change. Note that performance can appear higher than in the full evaluation (100 cases), as these top-ranked predictions are typically easier to explain due to more pronounced features.
>
> #### Table 1: D
>
> | λ1      | λ2 = 0        | λ2 = λ1 / 100 | λ2 = λ1 / 10 |
> |---------|---------------|---------------|--------------|
> | **0.01** | 6.24 \| 7.06  | 10.34 \| 7.94 | 5.95 \| 8.27 |
> | **0.1**  | 9.13 \| 8.44  | 9.91 \| 7.81  | 6.51 \| 6.82 |
> | **0.5**  | 10.75 \| 8.83 | 9.05 \| 6.89  | 8.09 \| 7.42 |
> | **1.0**  | 12.92 \| 8.42 | 13.64 \| 7.19 | 10.99 \| 8.48 |
> | **10.0** | 11.24 \| 6.81 | 8.50 \| 7.55  | 8.69 \| 7.30 |
>
>
> #### Table 2: FMA
>
> | λ1      | λ2 = 0        | λ2 = λ1 / 100 | λ2 = λ1 / 10 |
> |---------|---------------|---------------|--------------|
> | **0.01** | 6.76 \| 7.76  | 6.01 \| 7.81  | 10.08 \| 7.48 |
> | **0.1**  | 7.78 \| 8.15  | 9.50 \| 8.95  | 9.68 \| 9.31  |
> | **0.5**  | 7.16 \| 6.83  | 10.47 \| 8.17 | 12.63 \| 9.49 |
> | **1.0**  | 8.10 \| 6.95  | 10.24 \| 9.15  | 11.83 \| 7.15 |
> | **10.0** | 11.74 \| 7.04 | 11.63 \| 6.56 | 11.43 \| 7.71 |
>
>
> #### Table 3: GB
>
> | λ1      | λ2 = 0        | λ2 = λ1 / 100 | λ2 = λ1 / 10 |
> |---------|---------------|---------------|--------------|
> | **0.01** | 5.86 \| 15.78 | 5.54 \| 15.58 | 8.91 \| 15.16 |
> | **0.1**  | 4.64 \| 15.22 | 5.85 \| 14.84 | 9.14 \| 16.04 |
> | **0.5**  | 6.05 \| 16.92 | 9.27 \| 17.02 | 8.59 \| 15.48 |
> | **1.0**  | 7.04 \| 16.15 | 8.31 \| 16.82 | 9.30 \| 14.62 |
> | **10.0** | 7.95 \| 8.31  | 10.99 \| 11.16| 9.86 \| 11.93 |
>
> Generally, if λ1 is not too large, decreasing it by 1 or 2 orders of magnitude (and λ2 in accordance) does not have a huge effect on the returned masks. Slower convergence could also be compensated for by increased iterations. In this ablation study though we fixed iterations at 200 as in the general sepsis task. Outside of the range λ1=10 and λ1=0.01, we noted a performance drop off when keeping iterations fixed at 200.
>
> We will add this to our manuscript as well.

---

> > ### Comment · Reviewer_AND9 · 2025-08-06
> >
> > Thanks for your response. Most of my concerns have been well addressed by the author's thorough responses. Therefore, I have improved my score.

---

> > > ### Author Response · Authors · 2025-08-06
> > >
> > > Thank you once again for your time! We're glad to hear that we were able to address most of your concerns. Please do reach out if there’s anything else we can clarify before the 8th!

---

### Official Review · Reviewer_tzvM · 2025-07-05

**Clarity:** 3
**Significance:** 3
**Originality:** 2
**Rating:** 4
**Confidence:** 4

**Summary:**

The paper presents Continask, a novel approach to address the black-box interpretation problem of irregular time series data, specifically focusing on the healthcare, where up to 90% of data may be missing. The authors propose an innovative Deletion perturbation method to quantify missingness importance, and to resolve the non-differentiability issue of perturbation, this paper employs a NeuroEvolution strategy. The approach is validated on synthetic and real-world sepsis prediction tasks.

**Questions:**

1. Are there other specific methods or optimizations that could be explored to improve the computational efficiency of NeuroEvolution?
2. Would it be possible to benchmark the performance of MLP-F by comparing it to TimeX (with the same perturbation setup, i.e., modifying only the observations)? A direct comparison could highlight the competitive advantages or limitations of the proposed approach.
3. How do different hyperparameters affect the quality of the time mask? An ablation study focused on hyperparameters would help clarify which factors are most influential in model performance.
4. Could the sepsis prediction case study be extended to 20 instances? Increasing the sample size would help strengthen the statistical significance and provide a more robust validation of the method.

**Ethical Concerns:**

["NO or VERY MINOR ethics concerns only"]

**Final Justification:**

Most of my concerns have been addressed and the contribution of this paper is clear.

**Limitations:**

1.The slower computational speed of NeuroEvolution compared to gradient descent significantly limits the scalability of the model, particularly in large-scale datasets. This is a key area for improvement in future work.
2.The impact of Deletion perturbation in scenarios with high missing data is not fully explored. It would be helpful to investigate how this method behaves in such conditions, especially with regard to the realism of the missing samples.

**Paper Formatting Concerns:**

No major formatting issues.

**Quality:**

3

**Strengths And Weaknesses:**

Strengths:
1. The paper introduces a post-hoc instance-level explanation framework for irregular time series, which is a valuable contribution, particularly in the healthcare field where interpretability is crucial.
2. The innovative NeuroEvolution approach effectively addresses the non-differentiability issue in optimization problems, a novel solution compared to traditional gradient descent methods.

Weaknesses:
1. While NeuroEvolution shows promise, its computational efficiency is a limitation. It is significantly slower than traditional gradient descent methods, making it impractical for larger datasets. This limitation confines the sepsis analysis to only two cases, which could be a major drawback for real-world applications.
2. The discussion on the reliability of the Deletion perturbation method in high-missing-data scenarios could be expanded. It is unclear how the method performs when realistic samples are missing, which is common in real-world healthcare data.

---

> ### Author Rebuttal · Authors · 2025-07-31
>
> We'd like to thank the reviewer for their time and thoughtful comments. We were delighted to hear they found our paper 'innovative' and a 'valuable contribution'. In the below, we hope to answer all raised questions.
>
> **Q** Alternative optimizers
>
> **R**: As several reviewers raised concerns about computational speed, we investigated potential bottlenecks in our code more thoroughly. We found that the main bottleneck for the sepsis task lies in the Neural CDE (NCDE) needing to recalculate cubic spline coefficients at every model interaction. This is particularly expensive for irregular data (i.e., data with gaps), which limits parallelization.
>
> While we noted this as a potential issue in the original manuscript, we had underestimated its impact. To better understand this, we trained a multi-time attention network (mTAN) on the same sepsis task. Since mTAN processes raw time series directly, model interactions are significantly faster. This resulted in a 70× speed-up, reducing a process that previously required ~2 hours with the NCDE model to under 2 minutes with mTAN. Importantly, we only changed the model we are explaining, not Contimask. Hence, the slow speeds were related to the NCDE model, not Contimask. This generally showcases that Contimask can explain predictions within a much more practical time frame. Any method that uses perturbations on raw time series (especially with gaps, such as irregular time series) will experience a severe slow-down when explaining NCDE models given the expensive recomputation of the subic spline coefficients.
>
> Below, we report the average change in model output for 100 sepsis-positive and 100 sepsis-negative cases as predicted by mTAN, rather than the NCDE model (higher is better). While the Del perturbation outperforms FMA and GB at explaining cases that are predicted as sepsis-positive, GB performs best explaining cases predicted as sepsis-negative. We will include this table in the final version of the manuscript.
>
> #### Table 1: sepsis mTAN
>
> |       | **Sepsis**      |                | **No-sepsis**    |                |
> |-------|-----------------|----------------|------------------|----------------|
> |       | Del Odds Change | Imp Odds Change| Del Odds Change  | Imp Odds Change|
> | GB    | 4.41            | 10.45          | 6.29             | 11.80          |
> | FMA   | 4.16            | 3.41           | 7.26             | 7.19           |
> | Del   | 7.88            | 2.26           | 4.57             | 5.90           |
>
> We also extended the evaluation of the sepsis task with the NCDE model to 25 sepsis-positive and 25 sepsis-negative cases. Please find the results in the Table below, which show the same trend: while Del performs strongest for cases predicted to develop sepsis, it performs weakest on cases predicted to not develop sepsis.
>
> #### Table 2: sepsis NCDE
> |       | **Sepsis**      |                | **No-sepsis**    |                |
> |-------|-----------------|----------------|------------------|----------------|
> |       | Del Odds Change | Imp Odds Change| Del Odds Change  | Imp Odds Change|
> | GB    | 0.25            | 0.26           | 4.01             | 4.05           |
> | FMA   | 0.26            | 0.23           | 5.15             | 5.16           |
> | Del   | 4.97            | 0.21           | 2.72             | 3.94           |
>
> All models were trained with λ1=1, and λ2=0 (ablation study below).
>
> This difference in performance between Del and FMA/GB for sepsis-positive and -negative cases highlights Contimask's unique capabilities. As raised also by reviewr ykqd, we conducted a qualitative analysis into the returned masks. Using the Del perturbation, Contimask oftentimes removes entire sensor streams (i.e. channels), which drastically reduces the probability (according to mTAN and NCDE) that a patient becomes septic.
>
> Interestingly, there seems to be a structural difference in the data between patients who later develop sepsis, and those who don't: patients who later become septic are up to 2.6 times more likely to have observations in the ALT, AST, and Troponin channel (blood markers used to identify/monitor sepsis). Using the Del perturbation, Contimask reveals that both mTAN and NCDE are both heavily influenced by data being observed in these channels and use this as a main discriminator of whether a patient will become septic. Of course, whether these channels are observed is decided by the treating physicians who decide which information to monitor. Thereby, the models do not actually provide the clincians with any meaningful information but oftentimes just use the clinicians judgement as a discriminator.
>
> For cases prediced as septic-negative, removing data does not increase the predicted probability of developing sepsis as strongly and value altering perturbations (i.e., FMA and GB) produce better results.
>
> We believe these findings highlight Contimask's unique capabilites to uncover the reliance of irregular time series on whether data was observed at all, especially if using the Del perturbation. In clinical settings, this can help to reveal that a trained model does not actually uncovered meaningful patterns to help the treating physicians.
>
> Further optimizations are possible in future work. Although we ultimately selected the EvoTorch library after an extensive comparison of open-source alternatives, it remains sub-optimal because a significant portion of the PGPE optimizer’s computation still runs on the CPU.
>
> Another alternative would be to revert to using a single Tensor over a small feedforward network and apply genetic algorithms as a gradient-free optimization framework. Here the Tensor would be binary. While this approach can be efficient for short time series with few channels, it becomes computationally expensive for longer time series with many channels, as the parameter count would grow linearly with both length and channel number.
>
> **Q:** Benchmark against TimeX
>
> **R:** Unfortunately, this is not directly possible. While TimeX, a baseline primarily developed for regular time series, could in principle be used to explain models such as mTAN, we were unable to reproduce its results with the provided repository. The original implementation relies on a graph-neural-network–based model for irregular time series, but links for data download are currently broken, and it is unclear how to train the model or integrate a model like mTAN with the explainer. In addition, TimeX does not perturb data in the classical sense: it requires access to model internals and inspects the latent space. For models such as NCDE, this is likely tricky given the CDE solve. Since Contimask only queries the model and does not require access to model internals, a direct comparison would not be entirely fair, but we agree that it would be interesting to see how access to model internals improves explainability.
>
>
> **Q:** Hyperparameter ablation study
>
> **R:** We have included an ablation study on λ1 and λ2 in the Tables below for Del, FMA, and GB for the sepsis task using mTAN. These are computed for 10 cases predicted as sepsis-positive. The first value corresponds to Del odds change, and the second to Val odds change. Please note that results are at times higher than what is reported across 100 sepsis-positive cases since the top 10 predictions are easier to explain than the top 100 predictions (more confident predictions usually have more pronounced features).
>
> #### Table 3: D
>
> | λ1      | λ2 = 0        | λ2 = λ1 / 100 | λ2 = λ1 / 10 |
> |---------|---------------|---------------|--------------|
> | **0.01** | 6.24 \| 7.06  | 10.34 \| 7.94 | 5.95 \| 8.27 |
> | **0.1**  | 9.13 \| 8.44  | 9.91 \| 7.81  | 6.51 \| 6.82 |
> | **0.5**  | 10.75 \| 8.83 | 9.05 \| 6.89  | 8.09 \| 7.42 |
> | **1.0**  | 12.92 \| 8.42 | 13.64 \| 7.19 | 10.99 \| 8.48 |
> | **10.0** | 11.24 \| 6.81 | 8.50 \| 7.55  | 8.69 \| 7.30 |
>
>
> #### Table 4: FMA
>
> | λ1      | λ2 = 0        | λ2 = λ1 / 100 | λ2 = λ1 / 10 |
> |---------|---------------|---------------|--------------|
> | **0.01** | 6.76 \| 7.76  | 6.01 \| 7.81  | 10.08 \| 7.48 |
> | **0.1**  | 7.78 \| 8.15  | 9.50 \| 8.95  | 9.68 \| 9.31  |
> | **0.5**  | 7.16 \| 6.83  | 10.47 \| 8.17 | 12.63 \| 9.49 |
> | **1.0**  | 8.10 \| 6.95  | 10.24 \| 9.15  | 11.83 \| 7.15 |
> | **10.0** | 11.74 \| 7.04 | 11.63 \| 6.56 | 11.43 \| 7.71 |
>
>
> #### Table 5: GB
>
> | λ1      | λ2 = 0        | λ2 = λ1 / 100 | λ2 = λ1 / 10 |
> |---------|---------------|---------------|--------------|
> | **0.01** | 5.86 \| 15.78 | 5.54 \| 15.58 | 8.91 \| 15.16 |
> | **0.1**  | 4.64 \| 15.22 | 5.85 \| 14.84 | 9.14 \| 16.04 |
> | **0.5**  | 6.05 \| 16.92 | 9.27 \| 17.02 | 8.59 \| 15.48 |
> | **1.0**  | 7.04 \| 16.15 | 8.31 \| 16.82 | 9.30 \| 14.62 |
> | **10.0** | 7.95 \| 8.31  | 10.99 \| 11.16| 9.86 \| 11.93 |
>
> ContiMask is generally robust to moderate hyper-parameter changes.
>
> In the sepsis prediction task, the mask must cover a fixed input proportion. Very small λ1 and λ2  values cause the area constraint to be ignored, while very large values slow convergence since the changes in model output are overpowered. For the sepsis prediction task, the model output is in the range of -15 to 15, and λ1=1 forms a good starting point. Adjusting λ1 and λ2  one to two orders of magnitude below the threshold yields little additional change.
>
> **Q:** The impact of Deletion perturbation in scenarios with high missing data
>
> **R:** The sepsis prediction task is actually a scenario with a high ratio of missing data. As outlined above, while certain channels, such as heart rate, are almost continuously observed, others (e.g., bloodtests such as AST, ALT, Troponin) are missing for a large fraction of patients. It is in these scenarios that Contimask with the Del perturbation is particularly effective at detecting when the mere presence of data influences the model’s predictions as shown by our findings related to the channels AST, ALT, Troponin in the sepsis task (see above).

---

> > ### Author Response · Authors · 2025-08-07
> >
> > Dear Reviewer, please let us know if you have any remaining concerns.
> >
> > In the above rebuttal, we aimed to demonstrate that Contimask is well capable of providing timely explanations, assuming the underlying model allows for fast interaction. By using a different, commonly used model for irregular time series, we achieved a 70× speedup. We also extended the sepsis task experiments to include two models and explained 250 predictions, compared to just two previously.
> >
> > Additionally, we incorporated a qualitative analysis to highlight the value of Contimask and the Deletion perturbation, and conducted an extensive hyperparameter ablation study.
> >
> > Please reach out if there’s anything else we can clarify before the extended deadline on the 8th!

---

> > > ### Comment · Reviewer_tzvM · 2025-08-08
> > >
> > > Some other comments have been added focusing on further improvement in the revised version. Generally speaking, the contribution of this study is clear and the rebuttal has also addressed the major concerns. I will update my score.

---

> > ### Comment · Reviewer_tzvM · 2025-08-08
> >
> > Thanks the authors for their detailed rebuttal with additional experimental results. Some of the concerns have been well addressed, while some issues remain unsolved.
> >
> > (1) For the computational efficiency issue, more scalable optimization approaches should be explored in future work.
> > (2) As argued that the comparison with TimeX could not be performed. However, it would still be beneficial to provide a more detailed discussion on how Contimask differs from and potentially complements other existing methods for explaining time series models.
> > (3) More discussions needs to be provided focusing on practical application of Contimask in real-world settings.

---

> > > ### Author Response · Authors · 2025-08-09
> > >
> > > Thank you for your time and feedback!
> > >
> > > As highlighted in our qualitative assessment of the sepsis task, the ideal explanation tool for irregular time series models would combine structural perturbations such as Del, which alter the data structure, with value-altering perturbations (e.g., FMA, GB, or learned perturbations). This qualitative assessment also illustrates how real-world applications of Contimasks might look. We agree that scalability is key for real-world applicability.
> > >
> > > In the final version of our manuscript, we will stress more clearly how Contimask, and the Del perturbation in particular, can complement value-altering strategies, and how scalable optimization techniques should be explored to further enhance real-world applicability.

---

### Decision · Program_Chairs · 2025-09-17

**Decision:**

Accept (poster)

**Comment:**

This paper study explainability for black-box models for irregular time series data. The authors propose a method for perturbing input time series while incorporating irregularity-specific features into the saliency map. They also propose a genetic algorithm to learn the saliency maps without access to model gradients, potentially increasing the applicability. In the experiments, the proposed method outperforms two baselines on three datasets. The paper was overall received well by reviewers, who found the problem important and new, the solution clever and interesting, and overall found the results convincing. However, they also noted some key limitations. The main drawback from each reviewer was that the proposed method is extremely slow. However, during the discussion phase, this was somewhat resolved as the slow runtimes were tied to the NCDE model. Using a faster model, the authors expanded the experiments significantly. This should be included in the revised paper. Reviewers also noted that some experimental details were lacking and the work needed a hyperparameter study, which the authors included during the discussion period. Finally, reviewers noted there were limited explainability baselines. This was also resolved during the discussion period, as there aren’t methods to directly apply in this case. It would also strengthen the paper to put some examples in the main paper and potentially conduct a small human study to further verify the quality of the saliency maps. Overall, the paper studies an interesting and understudied problem with potential for the community to build on it.